# Usefulness of Dermoscopy in Localized Scleroderma (LoS, Morphea) Diagnosis and Assessment-Monocentric Cross-Sectional Study

**DOI:** 10.3390/jcm11030764

**Published:** 2022-01-30

**Authors:** Paulina Szczepanik-Kułak, Anna Michalak-Stoma, Dorota Krasowska

**Affiliations:** Department of Dermatology, Venerology and Paediatric Dermatology, Medical University of Lublin, 20-081 Lublin, Poland; annamichalak@wp.pl (A.M.-S.); dor.krasowska@gmail.com (D.K.)

**Keywords:** morphea, localized scleroderma, dermoscopy

## Abstract

Morphea, also known as localized scleroderma (LoS), is a chronic autoimmune disease of the connective tissue. The clinical picture of LoS distinguishes between active and inactive lesions. Sometimes the clinical findings are challenging to identify, and therefore, the need for additional methods is emphasized. Our study aimed to demonstrate the characteristic dermoscopic features in morphea skin lesions, focusing on demonstrating features in active and inactive lesions. In our patients (*n* = 31) with histopathologically proven LoS, we performed clinical evaluation of lesions (*n* = 162): active/inactive and according to both disease activity (modified localized scleroderma severity index, mLoSSI) and damage (localized scleroderma skin damage index, LoSDI) parameters. In addition, we took into account compression locations to determine whether skin trauma, a known etiopathogenetic factor in LoS, affects the dermoscopic pattern of the lesions. We performed a dermoscopy of the lesions, categorizing the images according to the severity within the observed field. We showed that within the active lesions (clinically and with high mLoSSI), white clouds and linear branching vessels had the highest severity. These features decreased within the observed field in inactive lesions and with high LoSDI. Brownish structureless areas were most intense in inactive lesions with high LoSDI. Erythematous areas, linear branching vessels, dotted vessels, and crystalline structures were statistically significant for pressure locations. We have shown dermoscopy is a valuable tool to assess the activity or inactivity of lesions, which translates into appropriate therapeutic decisions and the possibility of monitoring the patient during and after therapy for possible relapse.

## 1. Introduction

Morphea, also known as localized scleroderma (LoS), is a relatively rare autoimmune disease of still not fully understood etiology. Genetic, epigenetic, and environmental factors are suggested to be involved in its etiopathogenesis [1]. Usually, the disease has a characteristic three-stage course, including consecutive phases: inflammation, fibrosis, and atrophy. The active phases include inflammation and fibrosis of the skin, followed by a phase of nonactive changes, i.e., damage and atrophy [2]. Clinically, erythematous lesions are initially observed, sometimes with an edematous component, which is histopathologically observed as a perivascular inflammatory infiltrate combined with lymphocytes, plasma cells, eosinophils, and monocytes within the reticular layer of the dermis and subcutaneous tissue. In addition, vascular changes like dilation or formation of new blood vessels are observed. As the fibrosis, defined as the accumulation of extracellular matrix components progresses, one can observe indurated and sclerotic, porcelain-like lesions, often surrounded by an erythematous or violet margin [3]. In histopathological examination, inflammatory cells and deposition of thick, tightly packed, and homogenized collagen bundles, which are positioned parallel to the dermal-epidermal junction, are present in these lesions. Over time, the inflammation within the lesions is diminished, and atrophy, sometimes involving deep tissues, is seen. A common finding is the hyperpigmentation of the basal layer of the epidermis, which is manifested as a local discoloration of the skin [1,4].

According to the German guideline that considers the degree of fibrosis, five main types of LoS are distinguished: localized (subtypes: plaque, guttate, atrophoderma of Pasini and Pierini), generalized (subtypes: generalized localized, disabling pansclerotic), linear (subtypes: linear localized, en coup de sabre, progressive facial hemiatrophy), as well as deep and mixed [1,5].

Excessive, uncontrolled disease activity may result in permanent deformity and functional impairment. These problems, in correlation with undeniable cosmetic defects, contribute to poor quality of life for patients with morphea [6]. Therefore, early diagnosis and management are both of utmost importance [7]. Clinical assessment of lesion severity using localized scleroderma cutaneous assessment tool (LoSCAT) is crucial in determining the treatment [1]. LoSCAT, which is a standard tool, represents a combination of two indices, the modified localized scleroderma severity index (mLoSSI) and the localized scleroderma skin damage index (LoSDI). mLoSSI assesses the activity or severity of the disease and takes into account the presence of erythema, induration of the skin, and appearance of a new lesion or enlargement of an existing one within the last month. The LoSDI considers atrophy of the skin and subcutaneous tissue as well as hyper-hypopigmentation [8,9]. 

The clinical picture, although varied, is usually significant enough to establish the diagnosis in the majority of cases [5]. Noteworthy, skin lesions in patients with morphea, whose variable morphology depends on the disease stage and the potential for coexistence with other dermatoses, demand careful differential diagnosis [10] (Table 1).

Further diagnostic workup should be undertaken, especially in the inflammatory and atrophic phases, as well as in atypical or discrete lesions [11]. According to recent reports, nearly half of morphea biopsies are non-diagnostic and require the cooperation of clinicians and pathologists to correctly identify the disease [12]. Therefore, it seems to us that it would be important to explore non-invasive methods that would make it possible to observe pathological phenomena in the skin, especially in an outpatient setting or in children.

Dermoscopy is a non-invasive, inexpensive, and accessible method that allows for the rapid diagnosis of skin lesions, also in an outpatient setting. It enables not only observation of pigmented structures of the skin but also identification, differential diagnosis, monitoring, and prognosis of many inflammatory dermatoses [13,14]. The use of dermoscopy in LoS has been described in a few publications so far [11,15,16,17,18,19,20,21,22]. Undoubtedly, it is a valuable tool, minimizing the requirement for a skin biopsy or precisely indicating the location of the diagnostic specimen [11]. Its application facilitates the diagnosis of subclinical or discrete lesions because of the clear dermoscopic-histopathologic correlation [11,19,20,21]. Moreover, this method enables differential diagnosis, especially with extragenital variants of lichen sclerosus [16,18,21,22]. There is also information about the possibility of using dermoscopy to assess the efficacy of the treatment of morphea [15]. However, there is still a lack of information on whether there are certain dermoscopic characteristics that can be considered as markers of the disease activity. Furthermore, it has not yet been determined whether there are correlations between the dermoscopic pattern and mLoSSI/LoSDI values.

Our study aimed to demonstrate the characteristic dermoscopic features in morphea skin lesions. We specifically focused on demonstrating features in active and inactive lesions, which would help to facilitate the therapeutic decision, as well as to monitor the patient during treatment and to follow-up for recurrence. In addition, we performed a clinical evaluation using LoSCAT in each patient to investigate further the correlation between dermoscopic features and activity/damage indices. Moreover, referring to the role of skin trauma and the activation of fibroblasts in the etiopathogenesis of morphea [22,23], we distinguished the locations prone to long-term pressure among the observed lesions in different areas of the body to evaluate whether the presence of a mechanical factor is reflected in the characteristic features of the dermoscopic picture.

## 2. Materials and Methods

We conducted an observational single-center study in which we included all patients with a histopathologically confirmed diagnosis of LoS treated between May 2017 and October 2021 in the Department of Dermatology, Venereology, and Pediatric Dermatology Medical University of Lublin. Each patient underwent clinical evaluation, and, initially, the investigators determined whether the lesion was active/inactive, then used activity/damage indices, namely mLoSSI and LoSDI, for each lesion. We carefully noted the anatomical location of the lesions, focusing on whether the area was exposed to mechanical pressure, distinguishing areas such as shoulder, armpit, subpectoral, lower abdomen, pelvic girdle, and groin (Figure 1).

Each patient was then subjected to a dermoscopic examination. Dermoscopy was performed in the contact-polarized mode using FotoFinder Dermoscope (FotoFinder System GmbH, Bad Birnbach, Germany). The selection of dermoscopic variables was based on the available literature on dermoscopy of LoS [11]. The subsequent step was the evaluation of the dermoscopic images by two investigators, including one highly experienced doctor with over 20 years of practice. The expression of a given dermoscopic feature within the observed field was ordered quantitatively: 0 represented the absence of the feature, 1 (0–25% of the field covered by features), 2 (26–50% of the field covered by features), 3 (51–75% of the field covered by features), 4 (76–100% of the field covered by features).

Then all eligible images were submitted to statistical analysis.

All the data presented in this article are anonymous and are not identifiable, and the patients gave a written informed consent to be photographed. 

### Statistical Analysis

The statistical analysis was performed using the Statistica version 10.0 software (StatSoft Inc., Tulsa, OK, USA) for Windows.

Nonparametric tests were used for analysis:-Mann–Whitney U test—to test the significance of differences in the expression of dermoscopic features within the observed area between inactive and active lesions and between the absence and presence of pressure.-Spearman’s rank correlation coefficient significance test—to test the correlation between disease activity index and tissue damage index and expression of features within the observed field.

A value of *p* < 0.05 was considered statistically significant. 

## 3. Results

### 3.1. Characteristics of the Study Population

Thirty-one subjects (28 women and 3 men) with a limited type, plaque subtype of morphea, ranging in age from 8 to 74 years (mean age of subjects was 48.3 ± 22.5 years and the median was 59 years), were analyzed. The subjects had a total of 162 lesions. The majority of lesions were found on the trunk, with 58.0% of all lesions. The upper limb was involved in 26.5% of all lesions, and the lower limb in 14.8% of all lesions. One lesion was found on the head. Mechanical pressure affected 30.9% of all lesions (Table 2).

### 3.2. Clinical Evaluation of Skin Involvement

The vast majority of lesions were active (inflammatory and inflammatory-sclerotic)—80.9% of all lesions. The remainder were inactive (atrophic)—19.1% of all lesions. The mLoSSI value ranged from 0 to 9. The mean mLoSSI was 4.7 ± 2.5, and the median was 5. The LoSDI value ranged from 0 to 9. The mean of mLoSDI was 2.3 ± 2.0, and the median was 2. 

### 3.3. Dermoscopic Findings

The number (*n*) and frequency (%) of features assigned to categories 0–4 are shown in Table 3.

Within the observed lesions, white clouds had the highest expression (mean 1.83 ± 1.10 and median 2), followed by brownish structureless areas (mean 1.77 ± 1.11 and median 2), and vascular lesions: linear branching vessels (mean 1.30 ± 1.28 and median 1), as well as dotted vessels (mean 1.11 ± 1.24 and median 1).

The findings mentioned above are shown in Figure 2.

### 3.4. Correlation of Dermoscopic Findings with Lesion Activity, mLoSSI, mLoSDI, and the Presence of Pressure

#### 3.4.1. The Dermoscopic Findings vs. Lesion Activity

Based on statistical analysis, we found significant differences in the expression level of dermoscopic features within the observed field between the inactive lesion and active lesion only for the dotted vessels (*p* = 0.0153) and white clouds (*p* = 0.0138) (Table 4).

#### 3.4.2. The Dermoscopic Findings vs. mLoSSI Value

Statistical analysis showed significant correlations with disease activity score (mLoSSI) for features: linear branching vessels (*p* = 0.0016), white clouds (*p* < 0.0001) and brownish reticular areas (*p* = 0.0230). Interestingly, a positive correlation was found for linear branching vessels and white clouds. This translated to the fact that in lesions with high mLoSSI, these features had higher expression within the observed field. In contrast, a negative correlation occurred for brown reticular areas. The more active the lesion was, i.e., it had a higher mLoSSI value, the less frequently we observed these findings (Table 5).

#### 3.4.3. The Dermoscopic Findings vs. LoSDI Value

We have demonstrated significant correlations with LoSDI for features: linear branching vessels (*p* = 0.0344), white clouds (*p* = 0.0002) and brownish structureless areas (*p* = 0.0007). It appeared that as LoSDI increased, linear branching vessels and white clouds had less expression, and brownish structureless areas appeared more (Table 6). 

#### 3.4.4. The Dermoscopic Findings vs. Pressure Location

We found that erythematous areas (*p* = 0.0026), linear branching vessels (*p* = 0.0050), dotted vessels (*p* = 0.0016), and crystalline structures (*p* = 0.0026) had significantly higher expression in pressure localizations (Table 7).

## 4. Discussion

Based on the available literature, it should be emphasized that dermoscopy in LoS is a valuable diagnostic method. Campione et al. described two women with LoS in whom dermoscopy was used to monitor the effectiveness of topical therapy, i.e., imiquimod 5% in cream [15]. It was found that after a 16-week course of treatment, there was a complete remission of symptoms such as fibrosis and neovascularization [15]. Toader et al. presented a retrospective analysis of dermoscopic images of 17 patients in LoS, in whom they demonstrated the presence of white bands of fibrosis and branching vessels arranged in a network [18]. Saceda-Corralo presented the trichoscopy findings of 2 patients with a linear type of LoS in the scalp (en coup de sabre) [19]. The examination showed loss of hair follicle orifices on a whitish background, as well as black dots, broken hairs, and pili tori. On the other hand, within the marginal part of the lesions, a vascular pattern (short, thick linear vessels and winding, branching vessels) was visualized [19]. Bhat et al. described the case of a 28-year-old woman in whom dermoscopy of a sclerotic, hyperpigmented lesion revealed the presence of a typical pattern of white bands of fibrosis and branching vessels [21]. The researchers emphasized the utility of dermoscopy in differentiating steroid telangiectasias and vessels in LoS. In addition, they presented the possibility of using dermoscopy of skin lesions in LoS in dark-skinned patients, which, due to the lack of a clearly defined vascular edge, may affect diagnostic difficulties [21]. Peña-Romero et al. suggested the use of dermoscopy as a method to help differentiate active lesions in LoS (erythematous) from inactive lesions (telangiectasias) [20]. Nóbrega et al. described a case of a 9-year-old girl in whom dermoscopic findings of lesions together with clinical data were used to establish the correct diagnosis (LoS coexisting with extragenital lichen sclerosus), avoiding biopsy [17]. The other three publications dealt with the comparison of dermoscopic images of LoS and lichen sclerosus [11,16,22].

Our study included the largest cohort of patients with limited type LoS who had dermoscopy of affected skin lesions. We carried out a clinical evaluation of the patients’ lesions, determining their activity or lack of it, as well as validated activity/damage indices. In addition, we determined the location of the lesions and focused on areas of skin exposed to prolonged pressure, considered an important etiological factor for lesions in LoS [24]. These features were not described in the previous reports.

It has been shown that the most common dermoscopic finding with the highest expression level that we observed in our patients were white clouds, i.e., small, opaque, poorly demarcated areas. The name of this sign was proposed by Errichetti et al., and at the histological level, it corresponds to a sclerosis of the dermis associated with increased deposition of thickened collagen fibers [11]. It is noteworthy that the name “white fibrotic beams” was used in the past [13,16,18,21]. Various researchers have reported that the sign of the white clouds is a valuable diagnostic indicator in differentiating between LoS lesions and the extragenital variant of lichen sclerosus [11,16,21,22]. In both dermatoses, inflammation, fibrosis, and atrophy occur. Moreover, they may coexist, also within the same lesion [25]. In lichen sclerosus lesions white or white-yellow patches are usually observed, and they are larger, better demarcated, and brighter compared to white clouds [11,22]. This is because of the differences in the localization of collagen abnormalities across the levels of different skin layers; that is, in LoS, they are located deeper, at the reticular layer, while in lichen sclerosus, they are more superficial in nature [11]. 

Interestingly, our results indicated that the mean, as well as median expression of white clouds within the observed area, was higher for the active, inflammatory-sclerotic lesion. Furthermore, by correlating the intensity of this feature with standardized activity/damage markers, we showed that as the mLoSSI increased, this feature within the field tended to become greater, whereas it decreased as the LoSDI increased. This would indicate the obvious possibility of regarding white clouds as markers of disease activity and the potential to use them as helpful features in the assessment of disease remission and eventual completion of active phase immunomodulatory treatment. This may be particularly relevant to the reports of Errichetti et al., who demonstrated a dermoscopic-histological relationship in patients with LoS [11]. These investigators also stated that clear overlapping features between the clinical phases of LoS and white clouds might also be seen in the early stages of the disease, which translates into the use of this symptom in the diagnosis of early lesions. 

The most common vascular pattern observed in our patient group was linear branching vessels, which correlated positively with the mLoSSI score and negatively with the LoSDI score. However, in lesions clinically graded as "active" by us, this dermoscopic feature did not attain a statistically significant status, which only emphasizes the need for some testing in LoS in addition to the physical examination. Based on our results, we may assume that this feature is another potential marker of disease activity, and its disappearance may be associated with the extinction of skin lesion activity. Moreover, we showed that body regions exposed to prolonged pressure presented this type of vessels statistically more frequently. Linear branching vessels develop because of dermal vasodilatation or neovascularization phenomena. Of note, Beergouder et al. identified these vessels crossing white clouds as typical dermoscopic features for LoS [26]. However, in the work of Errichetti et al., this type of vessel at none of the clinical stages of the disease was determined to be statistically significant, in contrast to the irregular linear vessels, which were more often associated with the inflammatory-sclerotic phase than with the atrophic phase [11].

Another interesting result was the demonstration that dotted vessels were observed significantly more often within the active lesion, as well as in the pressure location, but this sign was not shown to be significant in terms of correlation with mLoSSI and LoSDI values. These vessels histologically correspond to dilated capillaries in regularly elongated dermal papillae and can be described as characteristic for psoriasis (vulgaris and pustular), lichen planus, pityriasis rosea, eczematous dermatitis, secondary lichenification, pityriasis rubra pilaris, or seborrheic dermatitis [27]. This symptom in the context of LoS has only been described by Errichetti et al. so far. They also observed similar vessels in patients with an extragenital variant of lichen sclerosus [11]. Undoubtedly, this is a phenomenon that requires further investigation.

The pigmentation of morphea foci is associated with the absence of disease activity, and its exact dermoscopic type had some clinical hallmarks. Brownish structureless areas were the most common feature among the pigmentation signs. We demonstrated that there was a positive correlation for LoSDI with this dermoscopic sign. However, in the case of mLoSSI, a negative correlation was found for the brownish reticular area. 

Interesting observations were made regarding pressure locations. Significantly more erythematous areas, linear branching vessels, dot-type vessels, and crystalline structures were observed in these areas. This may indicate that the most recent inflammatory foci of the disease were present in these areas. In addition, constant stimulation of the skin in this area is likely to exacerbate existing lesions.

Our study has some limitations that must be considered. First, no correlation was made between the dermoscopic features and histopathology; however, biopsy sites were included in the study. Second, we considered only the most common type of LoS---the limited type, plaque subtype. Furthermore, our cohort included mostly women and adults, and the vast majority of the images analyzed were active lesions. Nonetheless, these are limitations due to the rarity of the disease and the intent to present all results obtained. We did not present the results individually because we did not see consistency in the patients’ disease patterns; namely, most of them had coexisting active and inactive lesions. Therefore, we plan further studies in this area in the near future, especially including blinding (e.g., comparison with healthy skin, scars, or extragenital lichen sclerosus). 

## 5. Conclusions

Based on the obtained results and available literature, we would like to emphasize the special role of dermoscopy in LoS. This method is inexpensive, accessible, and easy to use, also in ambulatory settings. Its use enables us to indicate the area where the diagnostic specimen should be taken and to establish the diagnosis in correlation with clinical data. Its use facilitates the diagnosis of subclinical or discrete lesions, as there is a clear dermoscopic-histopathological correlation. Dermoscopy allows for differential diagnosis of LoS lesions, especially with the extragenital variety of lichen sclerosus. Moreover, as we have shown, it is a valuable tool to assess the activity or inactivity of lesions, which translates into appropriate therapeutic decisions and the possibility of monitoring the patient during and after therapy for possible relapse.

## Figures and Tables

**Figure 1 jcm-11-00764-f001:**
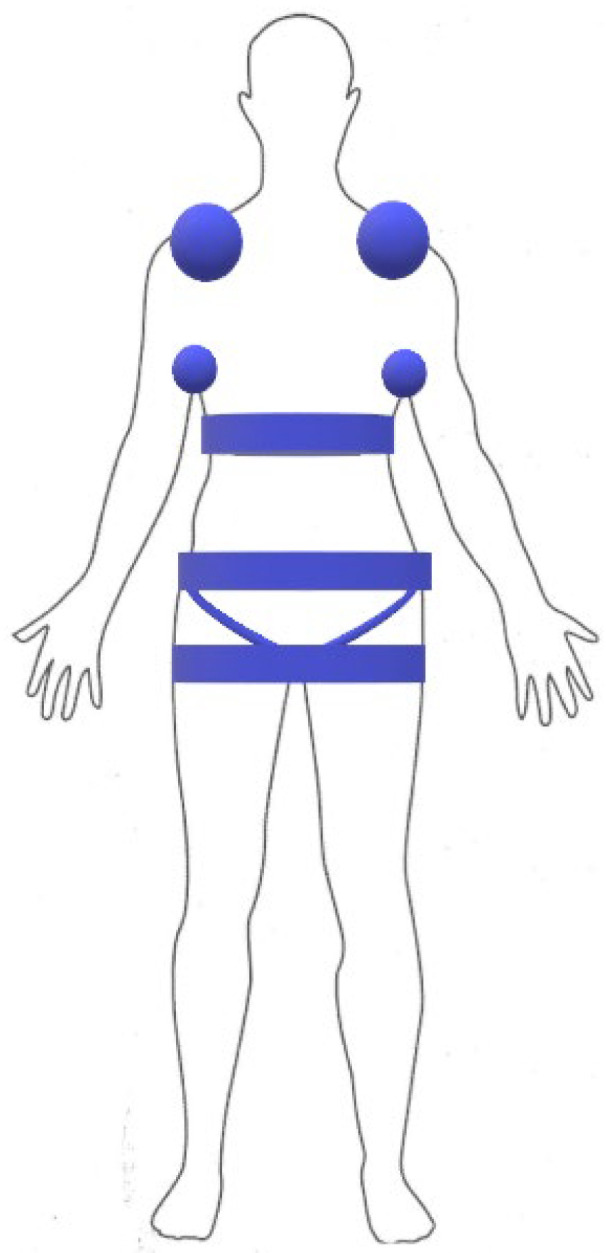
Compression locations considered.

**Figure 2 jcm-11-00764-f002:**
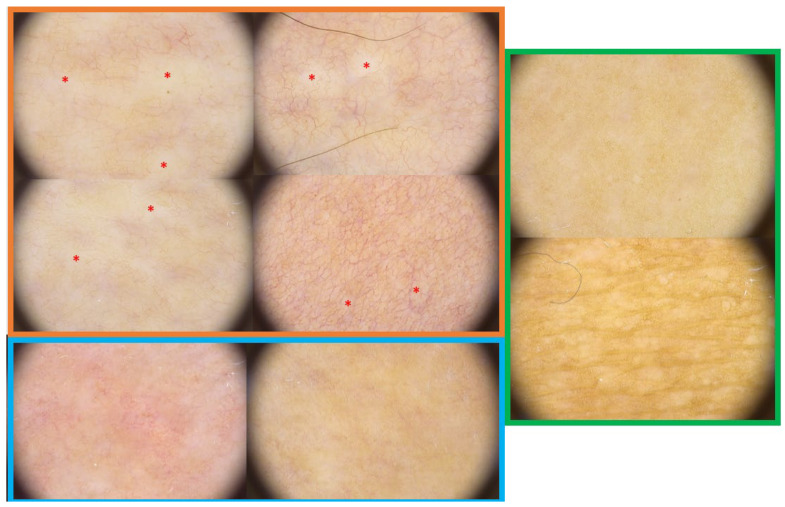
The orange box shows 4 dermoscopic images of active LoS lesions. The red asterisks mark the white clouds. Linear branching vessels are also seen scattered throughout the field. The blue box shows two dermoscopic images of active LoS lesions. Dotted vessels are visible, as well as widely scattered fine white clouds. The green frame shows a dermoscopic image of inactive lesions. The presence of reticulated brown areas is notable.

**Table 1 jcm-11-00764-t001:** Differential diagnosis of LoS.

LoS Differential Diagnosis	
LoS Phase	Skin Diseases
Early lesions: inflammatory and sclerotic	Mycosis fungoides (early lesions) Lichen sclerosus (early lesions) Stasis dermatitis Radiation dermatitis Vascular malformations in children Necrobiosis lipoidica diabeticorum
Predominantly fibrotic lesions	Systemic sclerosis Cutaneous sclerosis at the injection site Nephrogenic systemic fibrosis Drug-induced scleroderma-like lesions (bleomycin, taxanes) Keloid Carcinoma en cuirasse Skin metastasis
Late lesions: atrophic	Mycosis fungoides (advanced lesions) Lichen sclerosus (advanced lesions) Acrodermatitis chronica atrophicans Vitiligo Lichen planus planopilaris (late-stage lesions)

**Table 2 jcm-11-00764-t002:** Characteristics of The Study Population.

Location of Lesions	*n*	%
Trunk	94	58.0
Upper limb	43	26.5
Lower limb	24	14.8
Head	1	0.6
Total	162	100.0
**Compression Localization**	** *n* **	**%**
No	112	69.1
Yes	50	30.9
Total	162	100.0

**Table 3 jcm-11-00764-t003:** Dermoscopic findings. Quantitative assessment of the expression of a selected dermoscopic finding within the observed field: 0 represented the absence of the finding, 1 (0–25% of the field covered by findings), 2 (26–50% of the field covered by findings), 3 (51–75% of the field covered by findings), 4 (76–100% of the field covered by findings).

Dermoscopic Findings	0	1	2	3	4	Dermoscopic Findings	*n*	Mean	Standard Deviation	Median	Min.	Max.
Erythematous areas	142 (87.7)	17 (10.5)	3 (1.9)	0 (0.0)	0 (0.0)	Erythematous areas	162	0.14	0.40	0	0	2
Linear branching vessels	69 (42.6)	18 (11.1)	36 (22.2)	36 (22.2)	3 (1.9)	Linear branching vessels	162	1.30	1.28	1	0	4
Linear irregular vessels	109 (67.3)	28 (17.3)	24 (14.8)	1 (0.6)	0 (0.0)	Linear irregular vessels	162	0.49	0.77	0	0	3
Dotted vessels	73 (45.1)	33 (20.4)	29 (17.9)	19 (11.7)	8 (4.9)	Dotted vessels	162	1.11	1.24	1	0	4
Large purple vessels	143 (88.3)	19 (11.7)	0 (0.0)	0 (0.0)	0 (0.0)	Large purple vessels	162	0.12	0.32	0	0	1
White clouds	14 (8.6)	61 (37.7)	36 (22.2)	40 (24.7)	11 (6.8)	White clouds	162	1.83	1.10	2	0	4
Crystalline structures	158 (97.5)	3 (1.9)	1 (0.6)	0 (0.0)	0 (0.0)	Crystalline structures	162	0.03	0.21	0	0	2
Structureless brownish areas	26 (16.0)	38 (23.5)	52 (32.1)	40 (24.7)	6 (3.7)	Structureless brownish areas	162	1.77	1.11	2	0	4
Reticular brownish areas	140 (86.4)	1 (0.6)	8 (4.9)	9 (5.6)	4 (2.5)	Reticular brownish areas	162	0.37	0.98	0	0	4
Brownish dots	135 (83.3)	19 (11.7)	8 (4.9)	0 (0.0)	0 (0.0)	Brownish dots	162	0.22	0.52	0	0	2

**Table 4 jcm-11-00764-t004:** The dermoscopic findings vs. lesion activity.

Dermoscopic Findings	Lesion Activity				Z	*p*
	Inactive *n* = 31		Active *n* = 131			
	Mean ± Standard Deviation	Median (Min.-Max.)	Mean ± Standard Deviation	Median (Min.-Max.)		
Erythematous areas	0.23 ± 0.50	0 (0–2)	0.12 ± 0.37	0 (0–2)	1.32	0.1877
Linear branching vessels	1.00 ± 1.29	0 (0–4)	1.37 ± 1.27	1 (0–4)	−1.47	0.1427
Linear irregular vessels	0.52 ± 0.81	0 (0–3)	0.48 ± 0.76	0 (0–2)	0.25	0.8052
Dotted vessels	0.65 ± 1.02	0 (0–3)	1.22 ± 1.27	1 (0–4)	−2.42	0.0153
Large purple vessels	0.13 ± 0.34	0 (0–1)	0.11 ± 0.32	0 (0–1)	0.22	0.8247
White clouds	1.42 ± 1.34	1 (0–4)	1.93 ± 1.02	2 (0–4)	−2.46	0.0138
Crystalline structures	0.06 ± 0.36	0 (0–2)	0.02 ± 0.15	0 (0–1)	0.32	0.7514
Structureless brownish areas	1.94 ± 1.36	2 (0–4)	1.73 ± 1.04	2 (0–3)	0.58	0.5625
Reticular brownish areas	0.61 ± 1.43	0 (0–4)	0.31 ± 0.84	0 (0–3)	0.73	0.4656
Brownish dots	0.32 ± 0.60	0 (0–2)	0.19 ± 0.50	0 (0–2)	1.47	0.1409

Quantitative assessment of the expression of a selected dermoscopic finding within the observed field: 0 represented the absence of the finding, 1 (0–25% of the field covered by findings), 2 (26–50% of the field covered by findings), 3 (51–75% of the field covered by findings), 4 (76–100% of the field covered by findings). Z—Mann–Whitney U test value, *p*—probability level. Red indicates statistically significant, *p* < 0.05.

**Table 5 jcm-11-00764-t005:** The dermoscopic findings vs. mLoSSI value.

Variables	*n*	Rs	*p*
mLoSSI & erythematous areas	162	0.058	0.4661
mLoSSI & linear branching vessels	162	0.246	0.0016
mLoSSI & linear irregular vessels	162	−0.099	0.2118
mLoSSI & dotted vessels	162	0.001	0.9867
mLoSSI & large purple vessels	162	0.022	0.7806
mLoSSI & white clouds	162	0.535	<0.0001
mLoSSI & crystalline structures	162	0.041	0.6085
mLoSSI & structureless brownish areas	162	−0.141	0.0734
mLoSSI & reticular brownish areas	162	−0.179	0.0230
mLoSSI & brownish dots	162	−0.124	0.1153

mLoSSI: a modified localized scleroderma severity index. Rs—the value of Spearman’s rank correlation coefficient, *p*—probability level. Red marks are statistically significant, *p* < 0.05.

**Table 6 jcm-11-00764-t006:** The dermoscopic findings vs. LoSDI value.

Variables	*n*	Rs	*p*
LoSDI & erythematous areas	162	−0.011	0.8944
LoSDI & linear branching vessels	162	−0.166	0.0344
LoSDI & linear irregular vessels	162	0.076	0.3337
LoSDI & dotted vessels	162	−0.001	0.9940
LoSDI & large purple vessels	162	0.065	0.4079
LoSDI & white clouds	162	−0.286	0.0002
LoSDI & crystalline structures	162	0.038	0.6315
LoSDI & structureless brownish areas	162	0.264	0.0007
LoSDI & reticular brownish areas	162	0.117	0.1384
LoSDI & brownish dots	162	0.153	0.0513

LoSDI: Localized scleroderma skin damage index. Rs—the value of Spearman’s rank correlation coefficient, *p*—probability level. Red marks are statistically significant, *p* < 0.05.

**Table 7 jcm-11-00764-t007:** The dermoscopic findings vs. pressure location.

Dermoscopic Findings	Pressure Location				Z	*p*
	No *n* = 112		Yes *n* = 50			
	Mean ± Standard Deviation	Median (Min.-Max.)	Mean ± Standard Deviation	Median (Min.-Max.)		
Erythematous areas	0.08 ± 0.30	0 (0–2)	0.28 ± 0.54	0 (0–2)	−3.01	0.0026
Linear branching vessels	1.10 ± 1.16	1 (0–3)	1.74 ± 1.41	2 (0–4)	−2.81	0.0050
Linear irregular vessels	0.53 ± 0.75	0 (0–2)	0.40 ± 0.81	0 (0–3)	1.53	0.1252
Dotted vessels	1.32 ± 1.30	1 (0–4)	0.64 ± 0.94	0 (0–3)	3.15	0.0016
Large purple vessels	0.12 ± 0.32	0 (0–1)	0.12 ± 0.33	0 (0–1)	−0.07	0.9455
White clouds	1.78 ± 1.05	1 (0–4)	1.96 ± 1.23	2 (0–4)	−1.09	0.2754
Crystalline structures	0.00 ± 0.00	0 (0–0)	0.10 ± 0.36	0 (0–2)	−3.01	0.0026
Structureless brownish areas	1.81 ± 1.13	2 (0–4)	1.66 ± 1.06	2 (0–4)	1.14	0.2535
Reticular brownish areas	0.38 ± 1.01	0 (0–4)	0.36 ± 0.94	0 (0–4)	−0.04	0.9684
Brownish dots	0.21 ± 0.53	0 (0–2)	0.22 ± 0.51	0 (0–2)	−0.25	0.8033

Red marks statistically significant, *p* < 0.05. Z—Mann–Whitney U test value, *p*—probability level.

## Data Availability

Data sharing is not applicable.

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
