# Peer review of "Usefulness of Dermoscopy in Localized Scleroderma (LoS, Morphea) Diagnosis and Assessment-Monocentric Cross-Sectional Study"

_jcm, 2022, doi:10.3390/jcm11030764_

Round 1

Reviewer 1 Report

The authors present an interesting study about dermatoscopic features of localized scleroderma (morphea) with regard to activity of lesions and the influence of chronic pressure, which could allow facilitation of treatment and monitoring. Although the total number of included patients is relatively low, given the rarity of disease, this study has a remarkable sample size for a monocentric study. The manuscript represents an important piece of work as the literature about the topic is sparse and mainly consists of small case series or case reports. The authors embed their findings into a more general scope of cutaneous inflammation and therefore, this article will be of interest for a dermatological audience.

The article includes a well written introduction and thoroughly describes materials and methods. The results part requires some improvement as outlined further on. The images are of sufficient quality and are coherent by themselves. The article includes an important review of available literature and includes citations on all major aspects discussed. The discussion is written very clearly and has a good structure.

However, I would suggest some modifications to further improve the manuscript.

Major remarks:

  • The abstract by itself is not self-explaining. Abbreviations such as mLoSSI and LoSDI must be explained in the abstract, the parameter of compression by itself is not self-explaining

  • In my point of view, figure 2 and table 4 are superfluous as they exactly reflect the aforementioned text.

  • The main results from your tables should be briefly explained in the text, to do so, please write whole sentences (e.g. page 6, line 160-166).

  • There is a misleading wording regarding statistical tests: page 7 line 177-180 and page 9 line 208-211. These calculations are used to test a hypothesis, they cannot “show” a difference.

  • page 9 line 199-202: please rephrase

  • please evaluate mentioning or adressing further limitations of your study, e.g.:

    • potential differences with age of the patients (children vs. elderly)

    • potential differences with sex (overwhelming majority of patients female in this study)

    • the small number of inactive lesions may have biased statistical analysis

    • the study design was unblinded (the investigators rated known LoS lesions, no comparison to healthy skin or scars or lichen sclerosus as comparison)

    • there is no information regarding intraindividuell consistency of the findings (do many lesions in the same patient all appear the same upon dermatoscopy?)

    • were biopsy sites included in the study? (the LoS cases were histologically confirmed before study inclusion)

Minor remarks:

- The title should include the nature of the study: “monocentric cross-sectional study

- I suppose that the tables do not adhere to JCM style requirements; moreover, I suppose that tables should be mentioned in the text or in brackets (e.g. page 4, line 105).

- introduction: the terms fibrosis and sclerosis should be clearly defined at one point; from the viewpoint of a dermatopathologist there is mainly sclerosis in morphea (that is, decreased number of fibroblasts with increased amount of collagen).

- introduction: it would be useful to introduce a short section why pressure locations do matter for LoS patients and how pressure could affect the disease

- page 10 line 227 : pili torti

Author Response

Dr. Emmanuel Andrès

Editor-in-Chief

Journal of Clinical Medicine

Lublin, January 28 2022

Re: [JCM] Manuscript ID: jcm-1577929

Dear Editor,

Thank You for the insightful review of our manuscript entitled: “Usefulness of dermoscopy in localized scleroderma (LoS, morphea) diagnosis and assessment- monocentric cross-sectional study”. We appreciate the detailed comments of the Reviewers and we have changed our manuscript accordingly to their suggestions, as indicated below in our point-by-point reply. We have referred to literature and reconstructed the discussion to improve the quality of our manuscript.

Please find enclosed our revised manuscript (file named “manuscript_V2”) in which it has been indicated which lines were deleted (strikethrough, red colour) or added/changed (underlined, blue colour).

We hope Reviewers and Editor will be satisfied with all our responses to the comments and the revisions for the original manuscript. We feel that the changes indicated by the Reviewers have greatly improved the manuscript and we appreciate Your consideration for publication in “The Journal of Clinical Medicine”.  

Yours sincerely,

Paulina Szczepanik-Kułak, MD

Department of Dermatology, Venereology and Paediatric Dermatology

Medical University of Lublin, Poland

E-mail: vpaulinav@gmail.com

Phone No.: +48 889 987 497.

Response to Comments from REVIEWER #1

Dear Reviewer,

We would like to thank You for careful and thorough reading of this manuscript and for the thoughtful comments and constructive suggestions, which help to improve the quality of this manuscript. As below, on behalf of my co-authors, I would like to clarify some of the points raised in this review.

Comment 1:

The abstract by itself is not self-explaining. Abbreviations such as mLoSSI and LoSDI must be explained in the abstract, the parameter of compression by itself is not self-explaining.

Response:

We greatly appreciate the Reviewer for this comment. As suggested by the Reviewer, we have revised the content of the abstract to include clarification of the abbreviations mLoSSI and LoSDI and to describe in more detail the importance of pressure in the pathogenesis of morphea.

Comment 2:

In my point of view, figure 2 and table 4 are superfluous as they exactly reflect the aforementioned text.

Response:

We would like to express our sincere thanks for this comment. As recommended, we have deleted from the manuscript the Table 4 and Figure 2.

Comment 3:

The main results from your tables should be briefly explained in the text, to do so, please write whole sentences (e.g. page 6, line 160-166).

Response:

We certainly agree with the Reviewer's opinion, so we have reworded this section of the manuscript.

Comment 4:

There is a misleading wording regarding statistical tests: page 7 line 177-180 and page 9 line 208-211. These calculations are used to test a hypothesis, they cannot “show” a difference.

Response:

We greatly appreciate this valuable comment. We have changed the description of the statistical test we used so that it is not questionable.

Comment 5:

Page 9 line 199-202: please rephrase

Response:

Thank You. We have completed the indicated step and rewritten this section of the manuscript.

Comment 6:

Please evaluate mentioning or addressing further limitations of your study, e.g.:

    • potential differences with age of the patients (children vs. elderly)
    • potential differences with sex (overwhelming majority of patients female in this study)
    • the small number of inactive lesions may have biased statistical analysis
    • the study design was unblinded (the investigators rated known LoS lesions, no comparison to healthy skin or scars or lichen sclerosus as comparison)
    • there is no information regarding intraindividuell consistency of the findings (do many lesions in the same patient all appear the same upon dermatoscopy?)
    • were biopsy sites included in the study? (the LoS cases were histologically confirmed before study inclusion)

Response:

Thank You very much for this valuable comment. We acknowledge that our study had a couple of limitations, some of which were due to the rarity of the disease, which did not allow us to compare pediatric and adult or female and male populations. In addition, we paid special attention to active lesions, which require the most intensive therapy because of the possible risk of complications. Undoubtedly, the other limitations mentioned by the reviewer (lack of blinding, lack of unitary approach) are important and encourage us to conduct further studies in this still unexplored direction. Regarding the last comment, we evaluated all lesions in patients, including those undergoing biopsy.    

Comment 7:

The title should include the nature of the study: “monocentric cross-sectional study”

Response:

Thank You for the suggestion to include that information. We have changed the title of the paper.

Comment 8:

I suppose that the tables do not adhere to JCM style requirements; moreover, I suppose that tables should be mentioned in the text or in brackets (e.g. page 4, line 105).

Response:

We are grateful for this essential comment. As suggested, we have adjusted the tables to the editorial requirements and put the titles in brackets.

Comment 9:

Introduction: the terms fibrosis and sclerosis should be clearly defined at one point; from the viewpoint of a dermatopathologist there is mainly sclerosis in morphea (that is, decreased number of fibroblasts with increased amount of collagen).

Response:

We would very much like to thank you for this esteemed point. We have highlighted the definition of dermal fibrosis and associated sclerosis in the revised manuscript.

Comment 10:

Introduction: it would be useful to introduce a short section why pressure locations do matter for LoS patients and how pressure could affect the disease

Response:

We highly appreciate this suggestion. In the revised “Introduction” we have included information on why compression serves an important role in the pathogenesis of LoS.

Comment 11:

Page 10 line 227 : pili torti

Response:

Thank you. We have rewritten this word as suggested.

Reviewer 2 Report

I have no suggestions, the manuscript is well written , clear in contents. The numbers of patients is statistically sufficient . There are of course limitations but already explained by the authors

Author Response

Dr. Emmanuel Andrès

Editor-in-Chief

Journal of Clinical Medicine

Lublin, January 28 2022

Re: [JCM] Manuscript ID: jcm-1577929

Dear Editor,

Thank You for the insightful review of our manuscript entitled: “Usefulness of dermoscopy in localized scleroderma (LoS, morphea) diagnosis and assessment- monocentric cross-sectional study”. We appreciate the detailed comments of the Reviewers and we have changed our manuscript accordingly to their suggestions, as indicated below in our point-by-point reply. We have referred to literature and reconstructed the discussion to improve the quality of our manuscript.

Please find enclosed our revised manuscript (file named “manuscript_V2”) in which it has been indicated which lines were deleted (strikethrough, red colour) or added/changed (underlined, blue colour).

We hope Reviewers and Editor will be satisfied with all our responses to the comments and the revisions for the original manuscript. We feel that the changes indicated by the Reviewers have greatly improved the manuscript and we appreciate Your consideration for publication in “The Journal of Clinical Medicine”.  

Yours sincerely,

Paulina Szczepanik-Kułak, MD

Department of Dermatology, Venereology and Paediatric Dermatology

Medical University of Lublin, Poland

E-mail: vpaulinav@gmail.com

Phone No.: +48 889 987 497.

Response to Comments from REVIEWER #2

Dear Reviewer,

We would like to thank you for careful and thorough reading of this manuscript.

We kindly thank You for this incredibly kind review.
